# Incidence of Candidemia Is Higher in COVID-19 versus Non-COVID-19 Patients, but Not Driven by Intrahospital Transmission

**DOI:** 10.3390/jof8030305

**Published:** 2022-03-16

**Authors:** Marina Machado, Agustín Estévez, Carlos Sánchez-Carrillo, Jesús Guinea, Pilar Escribano, Roberto Alonso, Maricela Valerio, Belén Padilla, Emilio Bouza, Patricia Muñoz

**Affiliations:** 1Clinical Microbiology and Infectious Diseases, Hospital General Universitario Gregorio Marañón, 28007 Madrid, Spain; cscarrillo@salud.madrid.org (C.S.-C.); jguineaortega@yahoo.es (J.G.); pilar.escribano.martos@gmail.com (P.E.); roberto.alonso@salud.madrid.org (R.A.); mavami_valerio@yahoo.com.mx (M.V.); belen.padilla@salud.madrid.org (B.P.); emilio.bouza@gmail.com (E.B.); pmunoz@hggm.es (P.M.); 2Instituto de Investigación Sanitaria Gregorio Marañón, 28007 Madrid, Spain; 3Centro de Investigación Biomédica en Red (CIBER) de Enfermedades Respiratorias-CIBERES (CB06/06/0058), 28029 Madrid, Spain; 4Medicine Department, Faculty of Medicine, Universidad Complutense de Madrid, 28040 Madrid, Spain

**Keywords:** COVID-19, candidemia, viral–fungal coinfection, epidemiology, risk factors, invasive fungal infection, invasive candidiasis

## Abstract

There is scarce information on the actual incidence of candidemia in COVID-19 patients. In addition, comparative studies of candidemia episodes in COVID-19 and non-COVID-19 patients are heterogeneous. Here, we assessed the real incidence, epidemiology, and etiology of candidemia in COVID-19 patients, and compared them with those without COVID-19 (2020 vs. 2019 and 2020, respectively). We also genotyped all *C. albicans*, *C. parapsilosis*, and *C. tropicalis* isolates (*n* = 88), causing candidemia in both groups, providing for the first time a genotypic characterization of isolates gathered in patients with either COVID-19 or non-COVID-19. Incidence of candidemia was higher in patients with COVID-19 than non-COVID-19 (4.73 vs. 0.85 per 1000 admissions; 3.22 vs. 1.14 per 10,000 days of stay). No substantial intergroup differences were found, including mortality. Genotyping proved the presence of a low number of patients involved in clusters, allowing us to rule out rampant patient-to-patient *Candida* transmission. The four patients, involved in two clusters, had catheter-related candidemia diagnosed in the first COVID-19 wave, which demonstrates breaches in catheter management policies occurring in such an overwhelming situation. In conclusion, the incidence of candidemia in patients with COVID-19 is significantly higher than in those without COVID-19. However, genotyping shows that this increase is not due to uncontrolled intrahospital transmission.

## 1. Introduction

Nosocomial infections increased in patients admitted with COVID-19, partially due to the high use of antibiotics, steroids, and immunomodulatory drugs and they have been associated with worse prognosis and substantial increase of the economic burden [1,2,3]. Unfortunately, studies have been mostly addressed to describe bacterial superinfections [4,5] and Coronavirus-Associated Pulmonary Aspergillosis (CAPA) [6,7,8,9].

Data on candidemia in patients with COVID-19 is generally restricted to case reports [10,11,12] or small series of patients from different geographic areas [13,14,15,16,17]. In addition, heterogenous information comparing candidemia episodes in patients with and without COVID-19 is available [14,18,19,20,21]. However, it is necessary to assess further the actual incidence and outcome of candidemia in COVID-19 patients and to analyze its possible triggers, e.g., use of corticosteroids or antimicrobial agents, or the detrimental stewardship of central venous catheter (CVC) care during the pandemic [17,20,22,23]. Furthermore, no study has specifically assessed the potential role of horizontal *Candida* patient-to-patient transmission due to patient overcrowding or the impact of unrestricted use of antifungals by means of comparing the group of patients with COVID-19 with the one of patients with non-COVID-19.

We compared the incidence and clinical characteristics of candidemia in patients with and without COVID-19, unifying in the latter group all 2019 and 2020 episodes to ensure the lack of bias due to the restrictions of hospital admissions during the pandemic situation.

We also assessed the potential increase of patient-to-patient *Candida* transmission by genotyping the strains as well as the rate of antifungal resistance, including the study of the activity of a new drug, ibrexafungerp.

## 2. Materials and Methods

A retrospective study was conducted between January 2019 and December 2020 at Hospital General Universitario Gregorio Marañón—a 1200-bed healthcare center serving an urban area of nearly 350,000 inhabitants in Madrid, Spain. During the study period, the hospital cared for over 6000 SARS-CoV-2 patients; the number of beds was increased to 1572 in hospitalization wards and to 135 in adult ICUs during the weeks with the highest pressure on hospital services (March to April 2020).

All hospitalized adult patients with candidemia were included in the study. Comparisons between patients with and without previous SARS-CoV-2 infection (clinical characteristics, incidence, and density of incidence of candidemia episodes) were carried out. These comparisons between both groups of patients were also carried out in those admitted to ICUs.

Timing of the COVID-19 waves in Spain, during the study period, was as follows: first wave, 9 March to 24 June 2020; second wave, 4 August to 1 December 2020; third wave, 4 December 2020 to 31 March 2021.

### 2.1. Definitions

An episode of candidemia was defined as at least one peripheral blood culture (BC) positive for *Candida* spp. Patients with positive BCs from samples exclusively drawn from the catheter were excluded. 

The presence of two different species in the same set of BCs in a single patient was considered a polyfungal episode.

Multiple episodes were considered when *Candida* spp. was isolated in BCs of a given patient that elapsed more than 30 days from the incident episode, provided that microbiological and clinical resolution of the initial episode was achieved.

Persistent candidemia was defined as one or more positive follow-up BCs obtained ≥ 5 days from incident BCs in patients receiving antifungal therapy [24].

Patients with COVID-19 were those with positive SARS-CoV-2 using the reverse transcription-polymerase chain reaction (RT-PCR) assay in respiratory samples (nasopharyngeal swab, tracheal aspirate, bronchial aspirate, or bronchoalveolar lavage fluid). Candidemia in COVID-19 patients was defined as those presenting candidemia during hospital admission for SARS-CoV-2 infection. The group of non-COVID-19 patients included patients with candidemia but no evidence of COVID-19 diagnosed at any time point throughout 2019 and 2020.

### 2.2. Clinical Data

Patient clinical data were retrospectively collected from the electronic medical records and transferred to an anonymized database for statistical analysis (IBM SPSS Statistics software package, v26.0, Armonk, NY, USA) following a preestablished data collection protocol. Demographic variables, pathological history, unit at which the patient was admitted at the time of candidemia diagnosis, risk factors for candidemia including catheter use, antifungal and antibiotic treatment, superinfections, and description of candidemia and clinical outcome were included.

### 2.3. Microbiological Procedures

Three sets of aerobic and anaerobic BCs, collected from different sites (peripheral veins +/− catheters in patients carrying CVC) were performed for each patient, in accordance with hospital practices. The volume of whole blood ranged between 5 and 10 mL per bottle. Bottles were incubated at 35 °C until they were either flagged as positive or incubated for up to five days in an automated system (BACTEC^TM^ Plus Aerobic/F and BACTEC^TM^ Anaerobic/F, Becton Dickinson).

*Candida* species were routinely identified by matrix-assisted laser desorption ionization–time of flight mass spectrometry (MALDI-TOF MS), using a MicroFlexLT benchtop mass spectrometer (Bruker Daltonics GmbH, Bremen, Germany) and confirmed by the amplification of the ITS1-5.8S-ITS2 region. In vitro antifungal susceptibilities to amphotericin B, fluconazole, voriconazole, posaconazole (Sigma-Aldrich, Madrid, Spain), isavuconazole (Basilea Pharmaceutica, Basel, Switzerland), micafungin (Astellas Pharma, Inc., Tokyo, Japan), anidulafungin (Pfizer Pharmaceutical Group, New York, NY, USA), and ibrexafungerp (Scynexis, Inc., Jersey City, NJ, USA) were assessed applying the European Committee on Antimicrobial Susceptibility Testing (EUCAST) E.Def 7.3.2 broth dilution method using tissue-treated plates (CELLSTAR, 655180; Greiner bio-one, Frickenhausen, Germany). *C. parapsilosis* ATCC 22019 and *C. krusei* ATCC 6258 were used as quality controls. Isolates were categorized as resistant, susceptible, wild type, or non-wild-type according to clinical breakpoints or tentative epidemiological cutoff values (ECOFFs) provided by EUCAST [25]. Since there are no available clinical breakpoints for isavuconazole and ibrexafungerp, we used previously proposed tentative ECOFFs or wild-type upper limits [26,27]. All *C. krusei* isolates were considered intrinsically resistant to fluconazole. Phenotypically resistant isolates were retested. The *fks1* and *fks2* genes were sequenced in anidulafungin- and/or micafungin-non-wild-type isolates.

Species-specific microsatellite markers were used to genotype *C. albicans* (CDC3, EF3, HIS3 CAI, CAIII, and CAVI), *C. parapsilosis* (CP1, CP4a, CP6, and B), and *C. tropicalis* (Ctrm1, Ctrm10, Ctrm12, Ctrm21, Ctrm24, and Ctrm28) [28]. Capillary electrophoresis using the 3130xl analyzer (Applied Biosystems-Life Technologies Corporation, Carlsbad, California, USA) was performed with the GeneScan ROX marker; electropherograms were analyzed with the aid of the GeneMapper^®^ v.4.0 software (Applied Biosystems-Life Technologies Corporation, California). Genetic relationships between genotypes were studied by a Minimum Spanning Tree using BioNumerics version 7.6 (Applied Maths, Sint-Martens-Latem, Belgium). Isolates were considered to have identical genotypes when showing the same alleles in all loci. An intraward cluster was defined as a group of ≥2 patients infected by an identical genotype admitted to the same ward within a period of 12 months [28].

As per clinical practice, when ordered by the attending physician, serum samples were processed to detect (1, 3)-β-D-glucan (BDG). The Fungitell diagnostic test was used (until 14 July 2019) following manufacturer’s instructions (Fungitell, Cape Cod International, Inc., Falmounth, MA, USA). We applied the cutoff value proposed by the manufacturer (positive ≥ 80 pg/mL). From 15 July 2019, the Wako β-glucan test was utilized (Fujifilm Wako Pure Chemical Corporation, Osaka, Japan) with a cut-off value of 11 pg/mL.

### 2.4. Statistical Analysis

The Shapiro–Wilk normality test was applied. As some of the variables did not have a normal distribution, nonparametric tests were performed. Categorical variables are presented as frequencies and percentages. Results for continuous variables are expressed as medians and interquartile range (IQR). To detect significant intergroup differences, the Mann–Whitney U test for continuous variables and χ^2^ or Fisher’s exact test (when at least one expected frequency in a fourfold table is less than five) for categorical variables were used. For the calculation of the overall comparative incidence between COVID-19 and non-COVID-19 candidemia episodes, we chose candidemia episodes per 1000 admissions and 10,000 days of stay in 2019 and 2020; incidences were compared by using Epidat 4.2 software package (Consellería de Sanidade, Xunta de Galicia, Spain).

### 2.5. Ethical Approval

The study was approved by the Research Ethics Committee with Medicines of Hospital General Universitario Gregorio Marañón (CEIm; number MICRO.HGUGM.2020-038) on 12 January 2021.

## 3. Results

### 3.1. Incidence of Candidemia in Patients with and without COVID-19

There were 47,048 admissions in 2019 and 42,444 in 2020 (6763 COVID-19 and 35,681 non-COVID-19 patients). Mean hospital stay was 14.7 and 7.5 days for patients with and without COVID-19 requiring admission, respectively.

Overall, 103 episodes of candidemia (101 patients) were detected—46 in 2019 and 57 in 2020. Thirty-two episodes occurred in patients with COVID-19 (31.1%) and 71 without COVID-19 (68.9%). Candidemia in COVID-19 patients occurred mainly during the first wave of the pandemic (20/32; 62.5%), 11 in the second wave (34.4%), and one episode in the third wave (3.1%). Non-COVID-19 patients with candidemia were distributed in 46 cases in 2019 and 25 in 2020. Comparisons between both populations are shown in Table 1.

Incidence of candidemia was 4.73 episodes per 1000 admissions in the COVID-19 group and 0.85 in the non-COVID-19 group (*p* < 0.001); likewise, incidence density per 10,000 days of stay was 3.22 and 1.14, respectively (*p* < 0.001).

### 3.2. Comparison between Candidemia Patients with and without COVID-19

There were no statistical differences with respect to age or gender between both groups. Patients with COVID-19 were more commonly admitted to the ICU (71.9% vs. 32.4%) and required more CVC (93.8% vs. 70.4%), more total parenteral nutrition (TPN) (100% vs. 64.8%), and more previous therapy with corticosteroids (84.4% vs. 40.8%) (*p* < 0.01) compared with patients without COVID-19. In contrast, patients without COVID-19 showed higher frequencies of liver disease (6.2% vs. 28.2%), gastrointestinal conditions (18.8% vs. 43.7%), admission in medical wards (15.6% vs. 47.9%), and previous abdominal surgery (9.4% vs. 35.2%) (*p* ≤ 0.01).

No significant differences were found regarding previous use of antibiotics, bloodstream infections (BSI) (31.2% vs. 19.7%, *p* = 0.20) or urinary tract infections (15.6% vs. 15.5%, *p* = 0.98). However, lower respiratory tract infection was more frequent in COVID-19 patients (34.4% vs. 11.3%, *p* < 0.01), probably due to ventilator-associated pneumonia. Cytomegalovirus reactivation showed a trend towards higher frequency in patients with COVID-19 (21.9% vs. 11.3%, *p* = 0.22).

Regarding the origin, catheter-related candidemia was more frequent in COVID-19 patients (81.2% vs. 60.6%, *p* = 0.03), whereas a urinary tract origin was more frequent in patients without COVID-19 (0% vs. 11.3%, *p* = 0.05). There was no difference between abdominal focus and primary origin of candidemia or the rate of persistent candidemia (15.6% vs. 11.3%, *p* = 0.54).

Patients with COVID-19 received echinocandins more frequently as a first-line therapy (81% vs. 47.9%) and less fluconazole than patients without COVID-19 (12.5% vs. 40.8%, *p* < 0.01). There were no statistically significant differences in mortality between the two groups in those patients who received echinocandins as first-line treatment (61.5% vs. 41.2%, *p* = 0.12). Median duration of antifungal therapy was 12.5 days (IQR 5.0–21.5) in patients with COVID-19 and 16 days (IQR 9.0–26) in patients without COVID-19 (*p* = 0.38).

Patients with COVID-19 had higher frequency of associated septic shock (43.8% vs. 21.1%, *p* = 0.02). No intergroup differences in overall mortality (62.5% vs. 46.5%, *p* = 0.13) and other candidemia-associated complications, such us thrombophlebitis, ocular impairment, or endocarditis, were observed. Median days of hospital stay of patients with candidemia and COVID-19 was significantly longer compared with patients without COVID-19 (50 (34.2–85.0 IQR) vs. 40 (19–59 IQR), *p* = 0.02).

### 3.3. Comparison between COVID-19 and Non-COVID-19 Patients with Candidemia in the Intensive Care Unit

Table 2 describes the comparison of episodes of candidemia between COVID-19 and non-COVID-19 patients admitted to the ICU. Out of the critically ill patients admitted to our hospital in 2020 (*n* = 3430), 389 (8.8%) required ICU admission due to COVID-19. On the other hand, there were 3486 admissions to ICU in 2019. Overall, 23/32 (71.9%) COVID-19 patients and 23/71 (32.4%) non-COVID-19 patients with candidemia were admitted to the ICU. Taking these figures into account, we found an incidence of 59.1 per 1000 admissions in patients admitted to the ICU due to COVID-19 vs. 3.5 per 1000 admissions in patients admitted to the ICU for other reasons (*p* < 0.01).

No age, gender, and risk factors differences in candidemia patients were observed. Critical non-COVID-19 patients with candidemia more frequently had diabetes, gastrointestinal or neurological diseases, or prior hemodialysis. Prevalence of catheter-related candidemia was similar in both groups (78.3% vs. 65.2%). We found no differences concerning persistent candidemia episodes, previous or concomitant infections, days with CVC, or days since admission to the ICU to the diagnosis of candidemia. There were no differences regarding candidemia-related complications. Mortality in critical patients with COVID-19 trended to be higher (73.9% vs. 60.9%).

### 3.4. Serological (1, 3)-β-D-Glucan Results

BDG was determined in 56/101 patients and was positive in 71.4% (40/56). There was a median of four days (2–7 IQR) between the diagnosis of candidemia and the first BDG determination and of three days (1.0–5.5 IQR) between the start of the antifungal therapy and the first BDG determination.

Sensitivity was 93.3% (14/15) with Fungitell assay and 63.4% (26/41) with Wako test. Using the optimal cutoff level of 7 pg/mL in the Wako test, as recommended by some authors [29], the sensitivity increased to 78% (32/41). Median BDG values were 798 pg/mL (129–1425 IQR) and 16 pg/mL (7.6–65.9 IQR) with the Fungitell and Wako tests, respectively. No differences in BDG-positive candidemia were found between COVID-19 and non-COVID-19 patients (61.1% vs. 76.3%, *p* = 0.24). There was no difference in the mortality of patients with positive or negative BDG measured with Wako test (34.6% vs. 46.7%, *p* = 0.45).

### 3.5. Involved Species and Antifungal Susceptibilities

One hundred and six isolates were studied (three episodes were polyfungal fungemias) in COVID-19 and non-COVID-19 patients. *C. albicans* was the most commonly found species (58%), followed by *C. parapsilosis* (15.2%), *C. glabrata* (11.4%), *C. tropicalis* (9.5%), *C. krusei* (5%), and *C. kefyr* (0.9%). No statistically significant differences in the epidemiology of species between both groups of patients were found.

Eight isolates were fluconazole-resistant (*C. krusei*, *n* = 5; *C. glabrata*, *n* = 2; *C. albicans*, *n* = 1); one *C. glabrata* isolate was also resistant to anidulafungin and micafungin and harbored the F708S mutation outside the HS1 of the *fks2* gene. Overall, the rate of fluconazole and echinocandin resistance was 7.8% and 1%, respectively. Azole resistance was mainly due to the presence of *C. krusei* isolates. Most resistant isolates (*n* = 7) came from patients without COVID-19; the only multiresistant *C. glabrata* isolate was from a 54-year-old male with COVID-19, who underwent multiple intra-abdominal surgeries (thoracoabdominal aneurysm surgery secondary to aortic dissection, necrosectomy for severe necrohemorrhagic pancreatitis, duodenal resections, double J catheter placement for obstructive uropathy, etc.) and multiple isolations from respiratory, intra-abdominal, and urinary samples. The aforementioned patient had been receiving empirical treatment with anidulafungin for the last 20 days prior the diagnosis of candidemia (breakthrough infection).

Ibrexafungerp showed in vitro activity against all the isolates tested (MIC_50_ = 0.06 mg/L, MIC_90_ = 0.5 mg/L, MICs range 0.016–1 mg/L); all isolates were considered ibrexafungerp wild-type, regardless of the presence of fluconazole or echinocandin resistance. MIC distributions of the drugs tested against the isolates are shown in Appendix A.

### 3.6. Candida Isolates Genotyping

Out of the 88 *C. albicans*, *C. parapsilosis,* and *C. tropicalis* isolates genotyped using species-specific microsatellite markers, we found two *C. albicans* intraward clusters involving two patients with COVID-19 (Figure 1). One intraward cluster involved two patients admitted to the ICU in whom candidemia was diagnosed in April 2020 (six days elapsed); the other intraward cluster involved two patients admitted to postsurgery ICU in April 2020 (seven days elapsed). All patients involved in intraward clusters had catheter-related candidemia, which suggested patient-to-patient transmission. We did not detect any intraward clusters in patients without COVID-19 in 2019 nor 2020. Furthermore, this type of *C. albicans* intraward clusters had remained undetected in the hospital since 2015. We did not find intraward clusters suggestive of patient-to-patient transmission for *C. parapsilosis* and *C. tropicalis* isolates.

## 4. Discussion

We demonstrated a higher incidence of candidemia in patients with COVID-19 that was not driven by *Candida* spp. patient-to-patient transmission. In addition, candidemia in COVID-19 patients was mainly catheter-related and occurred in critically ill patients, but it did not lead to increased mortality.

Compared with previous reports, we present one of the largest series of candidemia in patients with COVID-19. Further to this, ours is the first study in which isolates causing candidemia during the COVID-19 pandemic have been genotyped to gain more insight about patient-to-patient transmission not only in patients with COVID-19 but also in patients with candidemia and non-COVID-19. Previous studies in which *Candida* genotyping was carried out are limited by the fact that the number of isolates was low, they were sourced exclusively from patients with COVID-19, and they were not designed to address the question of clonal spreading as a driver of higher candidemia incidence in COVID-19 patients [30,31]. As a whole, comparison of this population with patients without COVID-19 provides valuable and thorough information not previously described, since comparisons of candidemia episodes including patients with and without COVID-19 have been only occasionally carried out [14,18,19,20,21].

Nosocomial infections have been described in admitted COVID-19 patients, particularly in critically ill subjects. Our institution reported that superinfections occurred in 44.6% of the ICU COVID-19 patients [32]. BSIs (particularly bacterial) are one of the most frequent types of infection in this setting, with a reported incidence rate of 47 episodes (95% confidence interval (CI) 35–63) per 1000 patient-days at risk [33].

Rates of invasive candidiasis in COVID-19, including candidemia, range from 0.7 to 23.5% [34] depending on the analyzed population. In this study, prevalence of candidemia in COVID-19 patients is 0.47% of all COVID-19 admissions (32/6763) and 5.9% in critically ill COVID-19 patients (23/389). A study from Spain reported a rate of 0.7% (7/989) of fungal superinfections complicating hospitalized patients with COVID-19; four episodes were caused by molds and three by *Candida* (candidemia, candiduria, and complicated intra-abdominal candidiasis) [4].

Few studies have compared the incidence of candidemia before and during the COVID-19 pandemic. A Brazilian study described an increased incidence of candidemia per 1000 admissions throughout the COVID-19 pandemic (1.54 in the prepandemic period–January 2019 to February 2020–and 7.44 in the pandemic–March to September 2020, *p* < 0.001). Among the candidemia episodes in the pandemic period, 36% occurred in COVID-19 patients. However, the authors also suggested that this increase may be due to a reduction in admissions [14]. Here, we also found an increase in the incidence of candidemia both by patient group per 1000 admissions (COVID-19 (0.85) vs. non-COVID-19 (4.73)) and by period time (2019 (0.97) vs. 2020 (1.32)). Therefore, we observe a considerable increase in the incidence of candidemia in critically-ill COVID-19 patients compared with non-COVID-19 patients (59.1 vs. 3.5 per 1000 ICU admissions, *p* < 0.01). This may be due to a bias in the number of patients analyzed and admissions in each group but it also highlights the relevance of candidemia in COVID-19 ICU patients, probably due to the catheter use, prone positions, and corticosteroids use. However, no apparent statistically significant differences in the classical risk factors for candidemia were found in our series.

Some authors have proposed the role of immunosuppression, such as administration of tocilizumab [35] or high doses of corticosteroids [22], as one of the risk factors for candidemia in patients with COVID-19. These findings were not confirmed in a USA tertiary care center analyzing ICU patients with and without COVID-19 infection [19]. Previous bacterial BSI was found to be an independent risk factor for candidemia in COVID-19 patients (31.5% vs. 3.2%) [36], possibly due to the increased use of antibiotics and the consequent impact on gut microbiome composition. In our study, COVID-19 patients with candidemia had increased use of CVC, TPN, previous corticosteroid therapy, and ICU admission. On the other hand, non-COVID-19 patients with candidemia had more previous abdominal surgery.

Due to additional complications when managing CVCs in the pandemic (e.g., prone position, prolonged hospital stays, work overload), the main origin of candidemia episodes of our COVID-19 cohort were CVCs. A group of experts from the *Gruppo Accessi Venosi Centrali a Lungo Termine* (GAVeCeLT) in Italy [37] suggested additional preventive measures in this group of patients, such as the use of peripherally inserted central catheters; the use of femoral access to minimize the risk of operator contamination by patient’s oral, nasal, and tracheal secretions during insertion; and the use of ultrasound guidance for the insertion of any central venous access. The fact that catheter-related candidemia was more frequent in patients with COVID-19, and that the four patients linked to intraward clusters had catheter-related candidemia, indicates poor catheter care during the first wave of the pandemic as a potential explanation for the soaring increase of candidemia in COVID-19 patients.

We also found an increase in all etiology catheter-related BSI in our center during the COVID-19 pandemic (1.89 in 2019 vs. 5.53 in 2020 per 1000 admissions) [38]. These data reinforce the importance of optimal catheter management to prevent healthcare-associated infections, preestablished diagnostic algorithms, and stewardship programs under current conditions [23,39].

COVID-19 affected the pattern of antifungal use in our center, with a significant increase in the use of echinocandins as first-line antifungal treatment, probably due to the management of candidemia in the ICU and care of septic shock patients. Despite unrestricted use of antifungal drugs in the COVID-19 pandemic, we did not detect increased resistance rates; in fact, most resistant isolates were sourced from patients without COVID-19. Fluconazole resistance was mainly driven by the presence of intrinsically resistant species (*C. krusei*), which is in line with previously reported epidemiology of resistance assessed in our hospital between 2007 and 2019 [40]. We found an echinocandin-resistant isolate from a patient with COVID-19 who had received anidulafungin; previous use of echinocandins may promote echinocandin resistance, as reported in a series of patients in our own hospital [41].

Although not a specific problem of COVID-19 patients, the recent landscape of candidemia reveals an increasing incidence of non-albicans *Candida* species, some showing intrinsic resistance to antifungals or potential to acquire it [42]. To overcome the fluconazole and/or echinocandin resistance problem, the development of new drugs is urgently needed. The triterpenoid ibrexafungerp, a new class of glucan synthase inhibitor currently being evaluated in various phase III trials, has shown excellent bioavailability after oral intake [43]; this drug may eventually be a good alternative when the use of azoles or echinocandins is contraindicated for any reason. We found that all isolates here tested, including those showing resistance to fluconazole and/or echinocandins, were considered ibrexafungerp wild type, which is in line with the high activity of this drug against isolates from blood and other clinical sources [27]. Clonal outbreaks of fluconazole-resistant *C. parapsilosis* isolates have been reported in different hospitals [44]. In the current pandemic situation, an increase in candidemia by multidrug strains such as *Candida auris* has been reported in some countries, perhaps due to the difficulties in preserving nosocomial infection control programs [17]. However, none of those situations seem to be the case in our institution.

We have been genotyping all consecutive *Candida* spp. isolates from blood cultures since 2007 to track the presence of intraward clusters (identical genotypes infecting different patients admitted to the same hospital unit) as a surrogate marker of patient-to-patient hospital transmission and poor infection control measures. In a previous publication, we reported the presence of a high number of clusters in patients admitted at the hospital between 2007 and 2010; after that period, a campaign to promote awareness on catheter-related infection was started, with a consequent reduction in clusters [45]. Later, there was an intraward cluster-free period between 2016 and 2019. We found that despite of the complicated patient management experienced in the hospital in 2020, a moderate number of patients with candidemia, all with COVID-19, were involved in clusters (4/101 (3.96%)). Consequently, clusters reemerged in 2020, although patient-to-patient transmission was not a main driver of the increased incidence of candidemia in COVID-19 patients (4/32 (12.5%)).

Our study also underlines the potential differences between the Fungitell and Wako BDG assays. Friedrich et al. previously reported lower sensitivity of BDG for the diagnosis of candidemia using Fungitell vs. Wako (86.7% vs. 42.5%, respectively) [46]; these data are in line with our observations of lower sensitivity values using the Wako test (93.3% using Fungitell vs. 63.4%), even with the optimal cutoff point (7.0 pg/mL) (78%). However, there were no differences in mortality in BDG-negative cases. In a previous report by our institution, prevalence of persistent negative BDG results in candidemic adult patients using the Fungitell assay was 17.6% [47]. It was usually associated with catheter-related episodes with early control of the source and removal of the CVC and to significantly lower mortality. Further exhaustive study is required to understand this.

Thirty-day mortality rate of candidemia in tertiary care hospitals has been reported to be around 38% in Europe [48]; in our cohort, the overall mortality rate was 22% (23/103), increasing to 32% in patients with COVID-19. Moreover, Kayaaslan et al. described higher rates of overall mortality in patients with COVID-19 compared with patients without COVID-19 (87.5% vs 67.9%, respectively) [20]. However, the authors found no differences in mortality between the two groups of patients.

Our work is mainly limited by the fact that it is a single-center study, with the pandemic peculiarities of our hospital in comparison with other healthcare centers. Moreover, it is hard to assess all possible factors that may have contributed to the increase in the number of cases of candidemia episodes in COVID-19 patients. On the contrary, our study is strengthened because we have been able to report and conduct a real-life analysis in a period of high patient care workload. Finally, genotyping allowed us to assess the real impact of horizontal transmission during the pandemic.

## 5. Conclusions

In conclusion, our study proves that the candidemia incidence increased in COVID-19 patients, and mainly in critically ill patients receiving TPN or corticosteroids. We also conclude that even though infection control policies and antimicrobial stewardship programs were considerably affected during the pandemic, it did not notably lead to an increase in the number of patient-to-patient cases of candidemia or the rate of antifungal resistance. Further studies are needed to improve this superinfection’s clinical management, and special attention should be put on the correct management of catheters.

## Figures and Tables

**Figure 1 jof-08-00305-f001:**
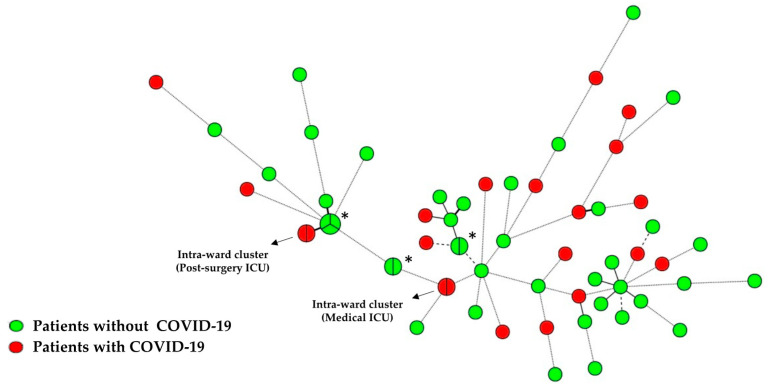
Minimum Spanning Tree showing the genetic relationship of *C. albicans* isolates. The source of the isolate is depicted in red (patients with COVID-19) or green (patients without COVID-19). Circles represent different genotypes, while partitions within the circles indicate the number of patients belonging to the same cluster. Connecting lines between the circles show profile similarities: solid bold, solid, dashed, and dotted lines indicate differences in one, two, three, or four markers, respectively. * Clusters involving patients with no epidemiological relationships are indicated by an asterisk.

**Table 1 jof-08-00305-t001:** Comparisons between COVID-19 and non-COVID-19 patients with candidemia.

Variables Studied	COVID-19 Patients *n* = 32 (31.1%)	Non-COVID-19 Patients *n* = 71 (68.9%)	*p*
Incidence per 1000 admissions (95% CI)	4.73 (3.24–6.67)	0.85 (0.67–1.08)	**<0.01**
Incidence density per 10,000 days of hospital stay	3.22 (2.20–4.50)	1.14 (0.90–1.40)	**<0.01**
(95% CI)			
Age—median (IQR)	65.5 (58.0–73.8)	65.0 (56.0–74.0)	0.9
Gender (male %)	23 (71.9)	44 (62.0)	0.33
Comorbidity			
Cardiovascular	15 (46.9)	39 (54.9)	0.45
Solid tumor	8 (25.0)	30 (42.3)	0.09
Neurologic disease	7 (21.9)	24 (33.8)	0.22
Diabetes mellitus	7 (21.9)	28 (39.4)	0.08
Gastrointestinal disease	6 (18.8)	31 (43.7)	**0.01**
Hemodialysis	6 (18.8)	14 (19.7)	1
Chronic kidney disease	5 (15.6)	21 (29.6)	0.13
Pulmonary disease	4 (12.5)	19 (26.8)	0.13
Liver disease	2 (6.2)	20 (28.2)	**0.01**
SOT recipients	2 (6.2)	9 (12.7)	0.49
Hematological malignancy	1 (3.1)	7 (9.9)	0.24
HIV	1 (3.1)	4 (5.6)	1
Hospital setting at candidemia diagnosis			
ICU	23 (71.9)	23 (32.4)	**<0.01**
Medical ward	5 (15.6)	34 (47.9)	**<0.01**
Surgical ward	4 (12.5)	14 (19.7)	0.57
Risk factors for candidemia			
Total parenteral nutrition	32 (100)	46 (64.8)	**<0.01**
Broad-spectrum antibiotics	31 (96.9)	66 (93.0)	0.43
Central venous catheter	30 (93.8)	50 (70.4)	**<0.01**
Corticosteroid therapy	27 (84.4)	29 (40.8)	**<0.01**
Previous ICU admission	25 (78.1)	17 (42.5)	**<0.01**
Previous colonization (six months)	22 (68.8)	35 (49.3)	0.06
Abdominal surgery	3 (9.4)	25 (35.2)	**<0.01**
Previous or concomitant infections			
Low respiratory tract infections (other than COVID-19)	11 (34.4)	8 (11.3)	**<0.01**
Bloodstream infection	10 (31.2)	14 (19.7)	0.20
CMV reactivation	7 (21.9)	8 (11.3)	0.22
Urinary tract infections	5 (15.6)	11 (15.5)	0.98
Other infections	5 (15.6)	11 (15.5)	0.98
Catheter-related candidemia	26 (81.2)	43 (60.6)	**0.03**
Persistent candidemia	5 (15.6)	8 (11.3)	0.54
Days with CVC previous candidemia, median (IQR)	18.0 (12.0–26.3)	16.5 (12.0–42.0)	0.54
First antifungal therapy			
Echinocandins	26 (81.2)	34 (47.9)	**<0.01**
Fluconazole	4 (12.5)	29 (40.8)	**<0.01**
Complications			
Septic shock	14 (43.8)	15 (21.1)	**0.02**
Thrombophlebitis	3 (9.4)	7 (9.9)	1
Ocular impairment	3 (9.4)	6 (8.5)	1
Outcome			
Overall mortality	20 (62.5)	33 (46.5)	0.13
Seven-day mortality	9 (28.1)	16 (22.5)	0.54
30-day mortality	19 (59.4)	29 (40.8)	0.08
Days from diagnosis of candidemia until death, median (IQR)	8 (4–23)	9.5 (4.0–20.0)	0.89
Hospital stay, median number of days (IQR)	50 (34.2–85)	40 (19–59)	**0.02**
*Candida* species *			
*Candida albicans*	22 (68.8)	40 (56.3)	0.23
*Candida tropicalis*	4 (12.5)	6 (8.5)	0.49
*Candida glabrata*	3 (9.4)	9 (12.7)	0.75
*Candida parapsilosis*	2 (6.2)	14 (19.7)	0.14
*Candida kefyr*	1 (3.1)	0	-
*Candida krusei*	0	5 (7.0)	-

CVC, central venous catheter; HIV, human immunodeficiency virus; ICU, intensive care unit; IQR, interquartile range; SOT, solid organ transplant; CI, confidence interval. *p* values presented in bold indicate statistical significance (*p* < 0.05). * Numbers and percentages are calculated over the number of isolates (*n* = 106).

**Table 2 jof-08-00305-t002:** Comparisons between COVID-19 and non-COVID-19 patients with candidemia admitted to the intensive care unit.

Variables Studied	COVID-19 Patients *n* = 23	Non-COVID-19 Patients *n* = 23	*p*
Incidence per 1000 admissions (95% CI)	59.1 (37.4–88.7)	3.5 (2.2–5.2)	**<0.01**
Age—median (IQR)	65 (57.7–74.2)	63 (54.5–70.0)	0.49
Gender (male %)	20 (87.0)	18 (78.3)	0.7
Comorbidity			
Cardiovascular	11 (47.8)	14 (60.9)	0.37
Solid tumor	6 (26.1)	5 (21.7)	0.73
Hemodialysis	6 (26.1)	13 (56.5)	**0.04**
Chronic kidney disease	4 (17.4)	10 (43.5)	0.06
Gastrointestinal disease	3 (13.0)	10 (43.5)	**0.02**
Diabetes mellitus	2 (8.7)	8 (34.8)	**0.03**
Liver disease	2 (8.7)	6 (26.1)	0.24
Neurologic disease	2 (8.7)	9 (39.1)	**0.02**
Pulmonary disease	2 (8.7)	5 (21.7)	0.41
SOT recipients	2 (8.7)	3 (13.0)	1
Hematological malignancy	1 (4.3)	4 (17.4)	0.35
HIV	1 (4.3)	1 (4.3)	1
Risk factors for candidemia			
Total parenteral nutrition	23 (100)	22 (95.7)	1
Central venous catheter	22 (95.7)	23 (100)	1
Corticosteroid therapy	22 (95.7)	17 (73.9)	0.09
Broad-spectrum antibiotics	22 (95.7)	23 (100)	1
Previous colonization (six months)	15 (65.2)	20 (87.0)	0.08
Abdominal surgery	2 (8.7)	5 (21.7)	0.41
Catheter-related candidemia	18 (78.3)	15 (65.2)	0.33
Persistent candidemia	4 (17.4)	4 (17.4)	1
Previous or concomitant infections			
Bloodstream infection	8 (34.8)	3 (13.0)	0.08
Ventilator-associated pneumonia	8 (34.8)	4 (17.4)	0.18
CMV reactivation	7 (30.4)	5 (21.7)	0.5
Days with CVC previous candidemia—median (IQR)	14 (11.2–19.5)	15.5 (11.2–22.0)	0.62
Days from ICU admission until candidemia episode, median (IQR)	19 (13.7–23.0)	16.5 (11.2–29.5)	0.99
Complications			
Septic shock	14 (60.9)	11 (47.8)	0.37
Ocular impairment	3 (13.0)	3 (13.0)	1
Thrombophlebitis	2 (8.7)	2 (8.7)	1
Outcome			
Overall mortality	17 (73.9)	14 (60.9)	0.34
Seven-day mortality	6 (26.1)	8 (34.8)	0.52
30-day mortality	16 (69.6)	11 (47.8)	0.13
Days from diagnosis of candidemia until death, median (IQR)	14 (4.5–24.5)	6.5 (3.0–39.7)	0.79
*Candida* species			
*Candida albicans*	17 (73.9)	12 (52.2)	0.13
*Candida non-albicans*	6 (26.1)	11 (47.8)	0.13

CMV, cytomegalovirus; CVC, central venous catheter; HIV, human immunodeficiency virus; ICU, intensive care unit; IQR, interquartile range; SOT, solid organ transplant. *p* values marked in bold indicate numbers that are significant (*p* < 0.05).

## Data Availability

Not applicable.

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
