# Peer review of "Incidence of Candidemia Is Higher in COVID-19 versus Non-COVID-19 Patients, but Not Driven by Intrahospital Transmission"

_jof, 2022, doi:10.3390/jof8030305_

Round 1

Reviewer 1 Report

The manuscript jof-1615604 entitled “Incidence of candidemia is higher in COVID-19 versus non-COVID-19 patients, but not driven by intra-hospital transmission” compare the incidence of candidemia with or without COVID-19 infection. They concluded that there was no significantly different mortality rate between these two groups. Furthermore, they also rule out intra-hospital Candida transmission,.

These data providing by the authors are interesting to those with closely related filed.

Since more COVID group using echinocandins as the first antifungal therapy, it would be interesting to compare whether there is different mortality rate between two groups using echinocandins as the first antifungal therapy. That is, the mortality rate of 28 in COVID-19 group vs 34 in non-COVID-19 group.

One minor point is that on line 69, the first and second waves of COVID-19 should be 2020 instead of 2021.

Author Response

Thank you for the interest generated in our manuscript, we hope that the revisions will bring more quality to this work in which we have put so much effort. These are the answers to your valuable suggestions.

Reviewer´s comment:  Since more COVID group using echinocandins as the first antifungal therapy, it would be interesting to compare whether there is different mortality rate between two groups using echinocandins as the first antifungal therapy. That is, the mortality rate of 28 in COVID-19 group vs 34 in non-COVID-19 group.

Authors’ reply: We have carried out such assessment and found no statistically significant differences (61.5% vs 41.2%, p=0.12). It has been added in the Results section (Lines 279-281).

Reviewer´s comment:  One minor point is that on line 69, the first and second waves of COVID-19 should be 2020 instead of 2021.

Authors’ reply: Thank you for spotting that mistake, we have corrected it.

Reviewer 2 Report

This is a well-written, interesting article that compares the epidemiology of Candida bloodstream infections in patients with and without COVID-19 revealing differences in incidence, risk factors and antifungal consumption. Intra-hospital Candida transmission was ruled out by genotyping the majority of the recovered isolates highlighting the need for heightened infection control measures particularly during the COVID-19 era, as the ones implemented in the authors’ center long before the pandemic.

Overall, the manuscript is clearly presented and easy to understand. I have several suggestions that may improve the clarity of the manuscript.

  1. “Claims of priority” as regards the statements “… providing for the first time a genotypic characterization of isolates in this situation” (lines 22-23), “… no study has analyzed the potential role of horizontal transmission …” (lines 47-48) and “… this is the first study in which isolates causing candidemia during the COVID-19 pandemic have been genotyped …” should be toned down.

In fact, Rajni et al. identified identical genotypes of C. tropicalis bloodstream isolates in ICU COVID-19 patients, suggesting patient-to-patient transmission (Open Forum Infect Dis. 2021, https://doi.org/10.1093/ofid/ofab452). Kordalewska et al. performed multilocus sequence typing of C. albicans isolates recovered from COVID-19 patients (J. Fungi 2021, https://doi.org/10.3390/jof7070552). Thomaz et al. reported an inter-hospital candidemia outbreak caused by clonal fluconazole-resistant C. parapsilosis isolates during the first year of the COVID-19 pandemic in Brazil (J. Fungi 2022, https://doi.org/10.3390/jof8020100). Arastehfar et al. reported fluconazole-resistant C. parapsilosis isolates that were clonal based on whole-genome sequencing and were recovered from ICU COVID-19 patients (Open Forum Infect Dis. 2022, https://doi.org/10.1093/ofid/ofac078). Of note, 2/4 papers were published last year.

Given the aforementioned, please rephrase and correct throughout the manuscript. The references provided should be acknowledged and commented in the discussion section.

  1. Line 69: Please correct the year of the first and second COVID-19 waves (2020 instead of 2021).
  2. Line 73-78: it is not clear by the definitions provided when a candidemic episode of a single patient was considered new, e.g. days of delay between the two episodes? blood culture clearance? resolution of clinical features of infection? Indeed, you mention 103 episodes and 101 patients, i.e. 2 patients had separate episodes, how were these defined?   
  3. Line 217: Given the impact of the pandemic particularly on the ICU environment caring for COVID-19 patients, I would suggest to present and comment the incidence of the infection in the COVID-19 ICU and non-COVID-19 ICU settings separately.
  4. Line 398: What is the rational for using the word “surprisingly”? Indeed, Friedrich et al. have reported that the sensitivity of the Fungitell assay (86.7%) is superior to that of the Wako β-glucan test (42.5%) for the diagnosis of candidemia (J Clin Microbiol. 2018, https://doi.org/10.1128/JCM.00464-18). Therefore, your results just confirm a previous finding. Please rephrase and include this information.
  5. Given the importance of gathering MIC distributions for different species-agent combination, particularly for isolates recovered from invasive infections, I would suggest to present them even as supplementary material.

Reviewer 3 Report

None

Author Response

Thank you for the interest generated in our manuscript, we hope that the revisions will bring more quality to this work in which we have put so much effort.